# Maternal intermittent fasting in mice disrupts the intestinal barrier leading to metabolic disorder in adult offspring

Yuan Liang[1,5], Wenzhen Yin [1,2,5], Chao Luo [1,5], Lijun Sun[1], Tiange Feng[1], Yunhua Zhang[1], Yue Yin [3✉] & Weizhen Zhang [1,4✉]

Maternal nutrition plays a critical role in energy metabolism of offspring. We aim to elucidate the effect of long-term intermittent fasting (IF) before pregnancy on health outcomes of offspring. Here we show long-term IF before pregnancy disrupts intestinal homeostasis of offspring with subsequent disorder of glucose and lipid metabolism. This occurs through the reduction in beneficial microbiota such as *Lactobacillus_intestinalis*. Our observations further support the concept that intestinal microbiota in offspring is vulnerable to maternal nutrition, and its homeostasis is critical for the integrity of intestinal barrier and metabolic homeostasis.

[1] Department of Physiology and Pathophysiology, School of Basic Medical Sciences, and Key Laboratory of Molecular Cardiovascular Science, Ministry of Education, Peking University, Beijing 100191, China. [2] Translational Research Center, Beijing Tsinghua Chang Gung Hospital, Beijing, China. [3] Department of Pharmacology, School of Basic Medical Sciences, and Key Laboratory of Molecular Cardiovascular Science, Ministry of Education, Peking University, Beijing 100191, China. [4] Department of Surgery, University of Michigan Medical Center, Ann Arbor, MI 48109-0346, USA. [5] These authors contributed equally: Yuan Liang, Wenzhen Yin, Chao Luo. ✉email: yueyin@bjmu.edu.cn; weizhenzhang@bjmu.edu.cn

Over the past two decades, intermittent fasting, not just simple calorie restriction, has emerged as an effective intervention for weight management and metabolic benefits[1,2]. Major IF strategies include alternate-day fasting (ADF), time-restricted feeding (TRF), alternate-day modified fasting (ADMF), 5:2 diet regimens, Ramadan fasting and other religious fasting[3]. In addition to large-scale human intervention studies, animal studies also confirm the metabolic benefits of intermittent fasting[4–6]. Mechanism underlying the metabolic benefits of intermittent fasting regimens remains largely unknown but may involve the gut microbiome, biological rhythm and lifestyle behaviors[1].

Despite the extensive studies focusing on the metabolic benefits of intermittent fasting, its effects on offspring health remain largely undefined. The intrauterine environment plays an important role in determining the health outcomes of the unborn fetus[7]. Poor environment due to maternal overnutrition or malnutrition during pregnancy may hamper fetal growth, development and health outcomes by permanently programming the fetus[8,9]. This concept is supported by several large epidemiological studies[10,11]. Maternal Ramadan exposure is associated with poorer health outcomes in offspring, including lower birth weight, increased incidence of type 2 diabetes and cardiovascular disease[12,13]. In animal studies, maternal low-protein diet or uterine artery ligation have been demonstrated to cause fetal intrauterine growth restriction, and higher risk of hyperglycemia, obesity, hypertriglyceridemia, and hypertension in adult offspring[14,15]. All these findings suggest that maternal malnutrition may increase the risk of metabolic disorders in offspring although its underlying mechanism remains unknown.

Intestine, the main organ for nutrient digestion and absorption, is susceptible to the maternal nutritional environment during intrauterine development[16,17]. Previous studies have shown that maternal nutritional restrictions during early to midgestation hinder intestinal development of offspring[18–20]. On the other hand, diet also plays an important role in host-gut microbiota interactions. Maternal microbiota is the most important microbiota source in the process of neonatal microbiota colonization[21,22]. Growing evidence shows that maternal nutritional status influences the microbiome composition and intestinal development of offspring[23], which may persist beyond birth and extend into adulthood[24,25]. Whether maternal intermittent fasting alters offspring intestinal homeostasis is currently unclear.

Here, we reported that long-term maternal IF before pregnancy disrupts the intestinal barrier of offspring by suppressing the beneficial microbiota such as *Lactobacillus_intestinalis*, leading to subsequent dysfunction in glucose and lipid metabolism.

## Results

**Metabolic benefits of intermittent fasting on dam**. There was moderate reduction in body weight and food intake in M-IF mice relevant to M-AL animals (Fig. S1a). Glucose tolerance was significantly improved in M-IF mice (Fig. S1b). Insulin tolerance and liver weight remained largely unaltered (Fig. S1c, d). Fat mass of perirenal white adipose tissue (rWAT), parametrial white adipose tissue (pWAT) and subcutaneous white adipose tissue (sWAT) were slightly reduced (Fig. S1e). These results suggest a metabolic benefit of intermittent fasting on dam.

**Maternal intermittent fasting deteriorates metabolism and intestinal barrier in offspring**. We next examined the metabolic consequence in O-IF offspring fed NCD. Food intake remained largely unaltered (Fig. S2a). Growth rate measured by change in body weight was slightly reduced without statistical significance (Fig. S2a). Significant impairment in glucose tolerance ($P = 0.0212$) (Fig. S2b) was observed in O-IF offspring, whereas insulin sensitivity remained unaltered (Fig. S2c). Liver weight ($P = 0.0373$), hepatic triglyceride

content ($P = 0.0072$), steatosis evidenced by Oil red O and H&E staining were elevated (Fig. S2d). Adipocyte size of eWAT was increased (Fig. S2e). Thus, maternal intermittent fasting deteriorates glucose and lipid metabolism in offspring.

Relevant to O-AL fed NCD, villus height was significantly decreased, whereas villus surface area increased in O-IF NCD mice (Fig. S3a). The stubby intestinal villus in O-IF offspring suggests an alteration in intestinal epithelial structure. We then examined gene relevant to tight junction. *Claudin-1(Cldn-1)* mRNA showed a slight reduction. Circulating lipopolysaccharides (LPS) showed no significant increase (Fig. S3b). Levels of TNF-*a*, IL-6 and IL-10 were slightly increased (Fig. S3c).

We next examined the effect of maternal intermittent fasting on offspring mice fed high fat diet (HFD). Relevant to ad libitum, intermittent fasting induced a profound alteration in intestinal barrier in offspring fed HFD. Villus height ($P = 0.0003$) was significantly decreased, whereas villus surface area ($P = 0.003$) and crypt depth ($P < 0.0001$) increased (Fig. 1a). The stubby intestinal villus in O-IF offspring suggests an alteration in intestinal epithelial structure. We then examined the intercellular tight junction, which is critical for intestinal barrier. Among the three major genes relevant to tight junction, *cldn-1* showed a significant reduction at mRNA ($P = 0.0417$) (Fig. 1b) and protein ($P = 0.0112$) (Fig. 1c) levels. The decrement in CLDN-1 was associated with substantial increase in circulating LPS (Fig. 1d), indicating disruption of intestinal barrier. In addition, mRNA levels of intestinal *Tnf-a* ($P < 0.0001$), *Ccl2* ($P = 0.0056$) and *Il-6* ($P = 0.0002$) increased significantly (Fig. 1e). Level of proliferating cell nuclear antigen (PCNA) in intestinal epithelium was significantly reduced ($P = 0.0127$) (Fig. 1f), whereas apoptosis was markedly increased as evidenced by TUNEL staining ($P = 0.008$) (Fig. 1g).

**Maternal intermittent fasting disrupts offspring intestinal microbiota characterized with a significant reduction in *Lactobacillus_intestinalis***. The intestinal mucosal barrier interacts with intestinal microbiota to maintain intestinal homeostasis. To investigate whether maternal intermittent fasting alters the intestinal microbiota of offspring, we performed 16 S rRNA sequencing to analyze the bacterial community structure in specimen of intestinal contents. The alpha diversity was significantly decreased in O-IF mice fed either NCD or HFD as indicated by Shannon index (Fig. 2a). Shown in Fig. 2b are the species with high relative abundance and their proportion at the species level in each group. Notably, relative abundance of *Lactobacillus_intestinalis* was decreased in O-IF mice fed either NCD or HFD (Fig. 2c). Compared to the O-AL NCD group, relative abundance of *Lactobacillus_intestinalis* was significantly decreased in O-AL mice fed HFD ($P = 0.0126$). To analyze the statistical differences in microbial communities between the O-AL and O-IF offspring, we compared OTUs with the LEfSe analysis. Species with relative abundance of taxons >10⁴ in LDA score were shown in Fig. 2d, including *g_Streptococcus, f_Streptococcaceae, g_Candidatus Arthromitus, f_unidentified Clostridiales, s_Lactobacillus_intestinalis, f_Lachnospiraceae* and *s_Streptococcus danieliae* for the O-AL offspring and *g_Methylocystis* for the O-IF animals. Cladograms (Fig. 2e) generated from the LEfSe analysis showed the most differentially abundant taxons enriched in O-IF offspring fed either NCD or HFD relative to O-AL offspring. Similarly, the O-AL offspring showed a greater abundance in the family *Lactobacillus*. All these results suggest a significant disruption in gut microbial community structure for the O-IF offspring fed either NCD or HFD. Among these bacteria, *Lactobacillus_intestinalis* is the dominant phylotypes contributed to the reduction of intestinal microbiota in O-IF offspring at the species level.

***Lactobacillus_intestinalis* restores the intestinal barrier dysfunction in offspring of maternal intermittent fasting mice**. Next, we investigated whether restoration of *Lactobacillus_intestinalis*

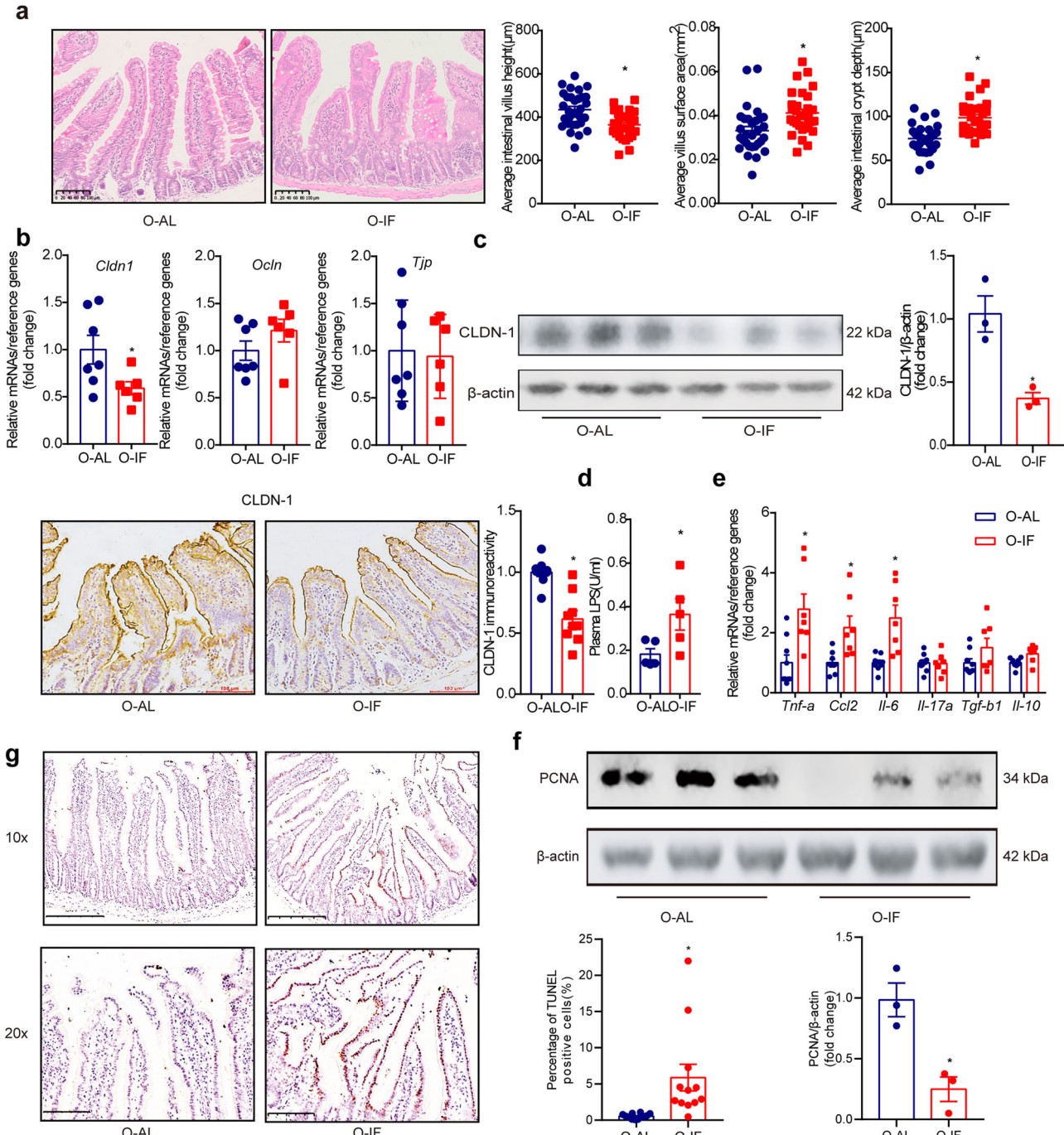

**Fig. 1 Maternal intermittent fasting disrupts intestinal barrier in offspring.** Four-week-old female mice were fed *ad libitum* (M-AL group) or alternate-day intermittent fasting (M-IF group) for 12 weeks, and resumed *ad libitum* after mating. Six-week-old offspring of M-AL group and M-IF group were fed high fat diet (HFD) for 12 weeks. Results were expressed as mean ± SEM. *$P < 0.05$ vs O-AL. **a** Intestinal histomorphology: H&E staining of the small intestine and quantitative results of villus height, villus surface area, and crypt depth. **b** mRNA levels of tight junction related genes (*Cldn1*, *Ocln* and *Tjp1*). These genes were determined by real-time quantitative PCR and normalized to the geometric mean value of reference genes (*Hprt*, *Rpl32* and *Tbp*). $N = 7$ and 6 for O-AL and O-IF respectively. **c** Protein levels of CLDN-1 in intestine. CLDN-1 was detected by Western blotting and immunohistochemical staining. The relative expression level was quantified using Image J software and normalized to β-actin. $N = 3$. **d** Plasma levels of LPS. $N = 5$. **e** mRNA levels of inflammation related genes (*Tnf-a*, *Ccl2*, *Il-6*, *Il-17a*, *Tgf-b1* and *Il-10*), which were determined by real-time quantitative PCR and normalized to the geometric mean value of reference genes (*Hprt*, *Rpl32* and *Tbp*). $N = 9$ and 7 for O-AL and O-IF respectively. **f** Levels of PCNA in intestine. The relative expression level of PCNA was quantified using Image J software. $N = 3$. **g** TUNEL staining of intestine.

(*L. intestinalis*) can rescue the phenotypes of intestinal barrier dysfunction in O-IF offspring fed HFD. Experimental design was shown in Fig. 3a. Administration of *L. intestinalis* restored the alteration in the intestinal villus evidenced by the increase of villus height ($P = 0.0002$) and concurrent decrease of villus surface area ($P = 0.0087$) in O-IF

offspring (Fig. 3b). Intestinal crypt depth remained largely unaltered (Fig. 3b). Further, treatment with *L. intestinalis* reversed the reduction in CLDN-1 ($P = 0.0273$) (Fig. 3c). Also reversed were the increases of plasma LPS ($P = 0.038$) (Fig. 3d) and *Tnf-a* ($P = 0.005$) (Fig. 3e). Other cytokines (Fig. 3e) showed only a marginal attenuation. Thus, *L.*

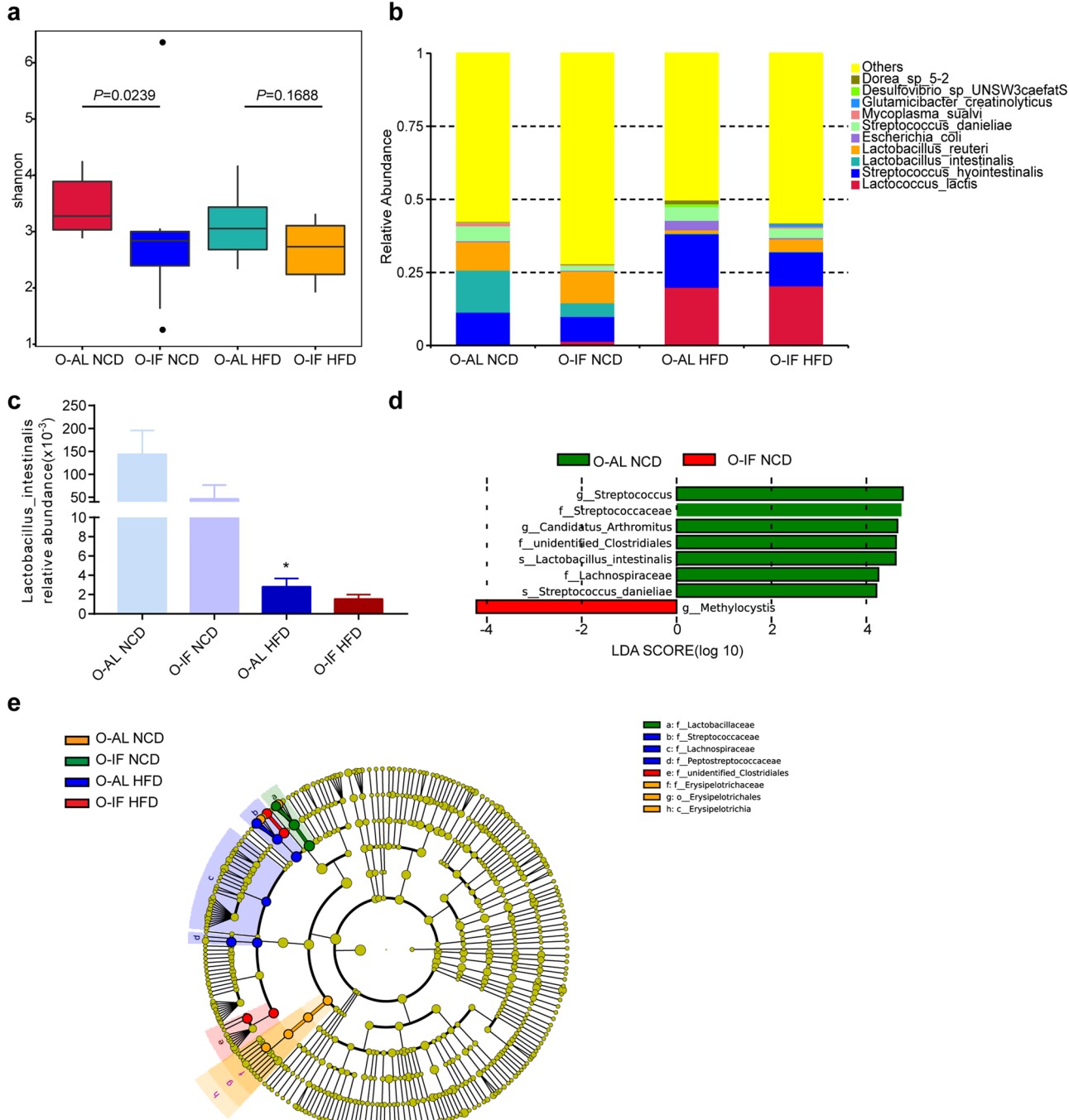

**Fig. 2 Maternal intermittent fasting disrupts offspring intestinal microbiota.** Six-week-old offspring of M-AL group and M-IF group were fed normal chow diet (NCD) or high fat diet (HFD) for 12 weeks. Intestinal microbiota was analyzed as described in the method section. $N = 9$ for O-AL NCD, 8 for O-IF NCD, 8 for O-AL HFD, and 10 for O-IF HFD. **a** Alpha-diversity-shannon diversity index of O-AL NCD, O-IF NCD, O-AL HFD and O-IF HFD, pairwise significance determined by Wilcoxon test. **b** The relative abundances of species in fecal samples. **c** Relative abundance of *Lactobacillus_intestinalis*. **d** The most differentially abundant taxons identified by LEfSe analysis with LDA score > 4. **e** Cladogram of significant changes at all taxonomic levels. The radiating circle represents the classification hierarchy from class to family. The size of node represents the abundance of taxa.

*intestinalis* restores the intestinal barrier dysfunction in offspring of maternal intermittent fasting mice.

**Lactobacillus_intestinalis improves the disorder of glucose and lipid metabolism in offspring of maternal intermittent fasting mice.** We next examined the metabolic consequence of intestinal barrier dysfunction in O-IF offspring fed HFD. Significant impairment in glucose tolerance (Fig. 4a) and increase in

circulating triglyceride ($P = 0.0196$) (Fig. 4b) were observed in O-IF offspring. Hepatic triglyceride content ($P = 0.0456$), and steatosis evidenced by H&E and oil red O staining were elevated (Fig. 4e). Fat mass and adipocyte size of perirenal white adipose tissue (rWAT), epididimal white adipose tissue (eWAT) and subcutaneous white adipose tissue (sWAT) were increased (Fig. 4f). Administration of *L. intestinalis* reversed the impairment in glucose tolerance (Fig.4a) as well as increment in plasma triglyceride (Fig. 4b), lipid absorption (Fig. 4c), intestinal *Cd36*

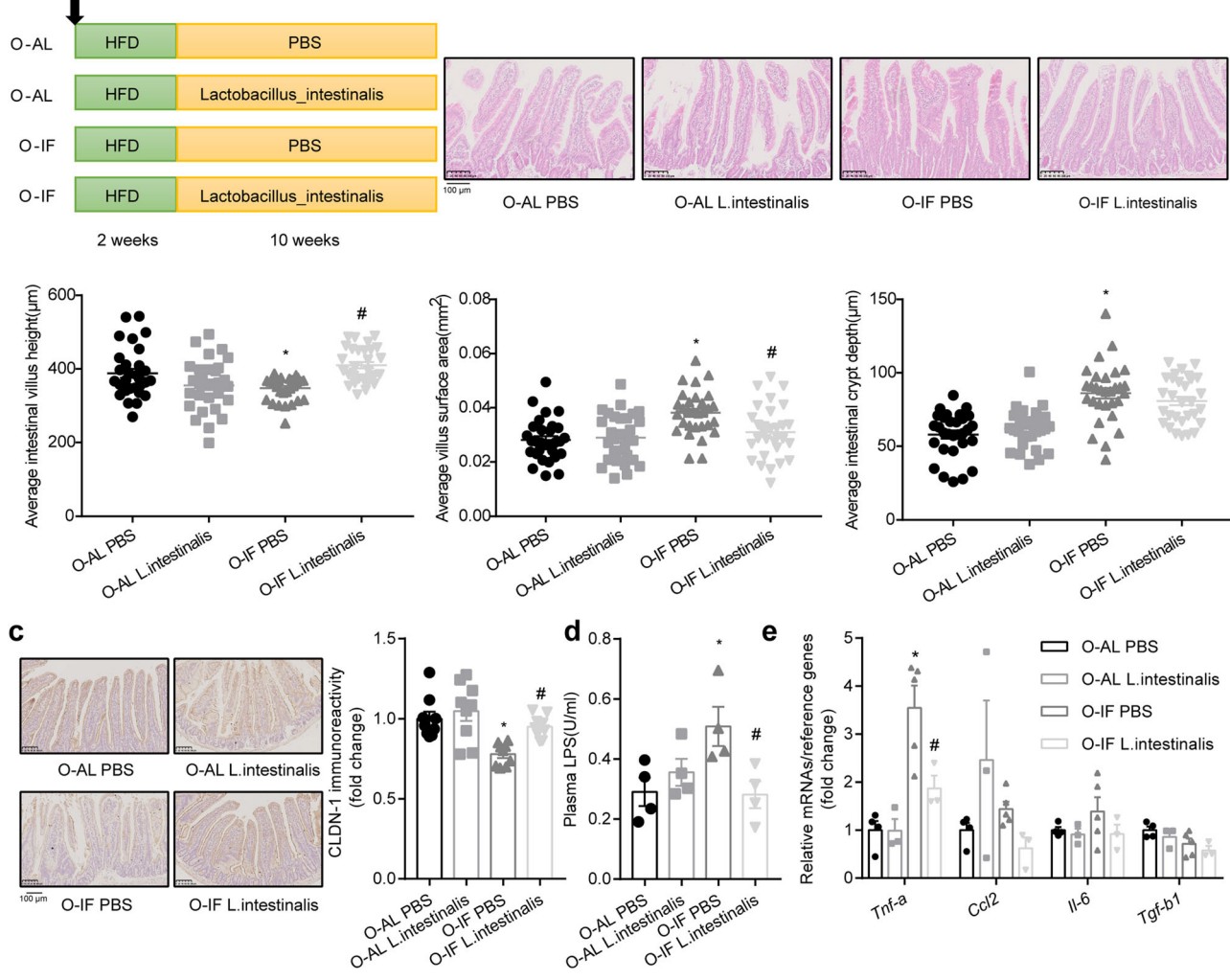

**Fig. 3 *Lactobacillus_intestinalis* restores the intestinal barrier dysfunction in offspring of maternal intermittent fasting mice. a** Experimental design. Six-week-old offspring of M-AL group and M-IF group were fed HFD for 2 weeks. PBS or *Lactobacillus_intestinalis* suspended in PBS at a dose of $2 \times 10^8$ CFU was administered by oral gavage once a day for 10 weeks. **b** Intestinal histomorphology. H&E staining of the small intestine and quantitative results of villus height, villus surface area, and crypt depth. **c** Immunoreactivity of CLDN-1. **d** Plasma levels of LPS. $N = 4$. **e** mRNA levels of inflammation related genes (*Tnf-a*, *Ccl2*, *Il-6* and *Tgf-b1*), which were determined by real-time quantitative PCR and normalized to the geometric mean value of reference genes (*Hprt*, *Rpl32* and *Tbp*). Results were expressed as mean ± SEM. *$P < 0.05$ vs. O-AL PBS. #$P < 0.05$ vs. O-IF PBS. $N = 4$ for O-AL PBS, 3 for O-AL *L. intestinalis*, 5 for O-IF PBS, and 3 for O-IF *L. intestinalis*.

($P = 0.0008$) (Fig. 4d), liver steatosis measured by oil red O and H&E staining (Fig. 4e), and adiposity (Fig. 4f).

**Offspring of maternal intermittent fasting mice co-housed with healthy mice restores intestinal barrier dysfunction.** As co-housed experiment has been demonstrated to be efficient for the transfer of intestinal microbiota, we used this approach to determine whether microbial transfer from healthy mice could rescue intestinal barrier dysfunction in maternal IF offspring fed HFD. Experimental design was outlined in Fig. 5a. After 10 weeks of co-housing, intestinal barrier was examined. O-IF co-housed with healthy mice showed a restoration in intestinal villus height ($P = 0.0175$) and surface area ($P < 0.0001$) (Fig. 5b), the decrement in *cldn-1* mRNA ($P = 0.0953$) and protein ($P < 0.0001$) (Fig. 5c) and increment in *Tnf-a* ($P = 0.0071$) (Fig. 5d). On the other hand, O-AL cohoused with O-IF developed intestinal barrier dysfunction evidenced by significant change of intestinal epithelial structure and CLDN-1 immunoreactivity in O-AL

cohoused relative to O-AL separated, indicating mutual transfer and of intestinal microbiota and its subsequent effect on intestinal barrier. Thus, a transferable component of the microbiota in heathy mice can rescue intestinal barrier dysfunction in maternal IF offspring.

**Offspring of maternal intermittent fasting mice co-housed with healthy mice shows improvement in glucose and lipid metabolism.** Co-housing of O-IF with healthy O-AL reversed the dysfunction in glucose and lipid metabolism (Fig. 6). Relevant to O-IF housed alone, O-IF co-housed with healthy mice showed an insignificant improvement in glucose tolerance (Fig. 6a), reduction in plasma triglyceride ($P = 0.0464$) (Fig. 6b), decreased lipid absorption (Fig. 6c), decrement of liver steatosis measured by oil red O and H&E staining (Fig. 6d), as well as adiposity evidenced by fat mass and adipocyte size of white adipose tissues (Fig. 6e). Meanwhile, healthy O-AL mice co-housed with O-IF demonstrated an impairment in glucose and lipid metabolism (Fig. 6).

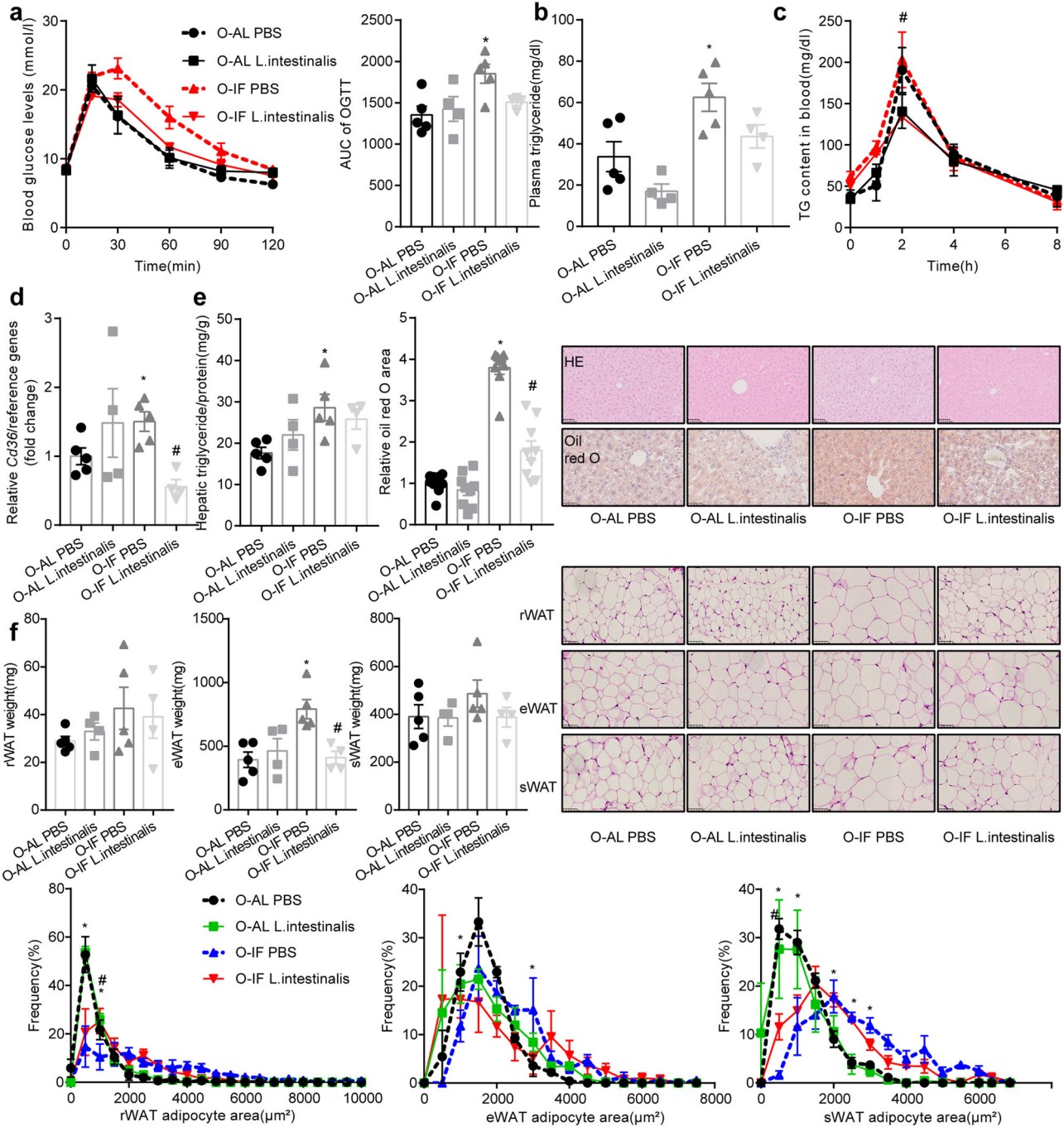

**Fig. 4 *Lactobacillus_intestinalis* improves glucose and lipid metabolism in offspring of maternal intermittent fasting mice.** *Lactobacillus_intestinalis* at a dose of $2 \times 10^8$ CFU/day were orally gavaged to O-AL or O-IF mice for 10 weeks. Results were expressed as mean ± SEM. Two-way ANOVA was used for comparisons between multiple groups. *$P < 0.05$ vs. O-AL PBS. #$P < 0.05$ vs. O-IF PBS. $N = 5$ for O-AL PBS, 4 for O-AL *L. intestinalis*, 5 for O-IF PBS, and 4 for O-IF *L. intestinalis*. **a** Glucose tolerance test and the area under curve. **b** Plasma TG levels. **c** Levels of circulating triglyceride in response to oral administration of olive oil. $N = 4$. **d** mRNA levels of *CD36*, which were determined by real-time quantitative PCR and normalized to the geometric mean value of reference genes (*Hprt, Rpl32* and *Tbp*). **e** Lipid contents and steatosis in liver. **f** Fat mass, H&E staining and adipocyte size in rWAT, eWAT and sWAT.

## Discussion

With increasing attention to metabolic health, IF has become a popular lifestyle choice. However, the influence of long-term maternal IF before pregnancy on the offspring remains largely unknown. Our studies indicate that prolonged maternal IF may disrupt intestinal barrier by reducing the beneficial microbiota with metabolic consequence in adult offspring (Fig. 7b). Alternate-day fasting for 12 weeks before pregnancy reduced

beneficial intestinal microbiota such as *L. intestinalis*, suppressed the expression of intestinal tight junction protein CLDN-1, impaired intestinal barrier evidenced by increased of circulating LPS and intestinal inflammatory molecules like TNF-a in adult offspring. The disruption in intestinal barrier was associated with a subsequent impaired glucose tolerance, increased hepatic steatosis and adiposity. Supplementation of *L. intestinalis* restored the intestinal barrier function and metabolic phenotypes. Cohousing

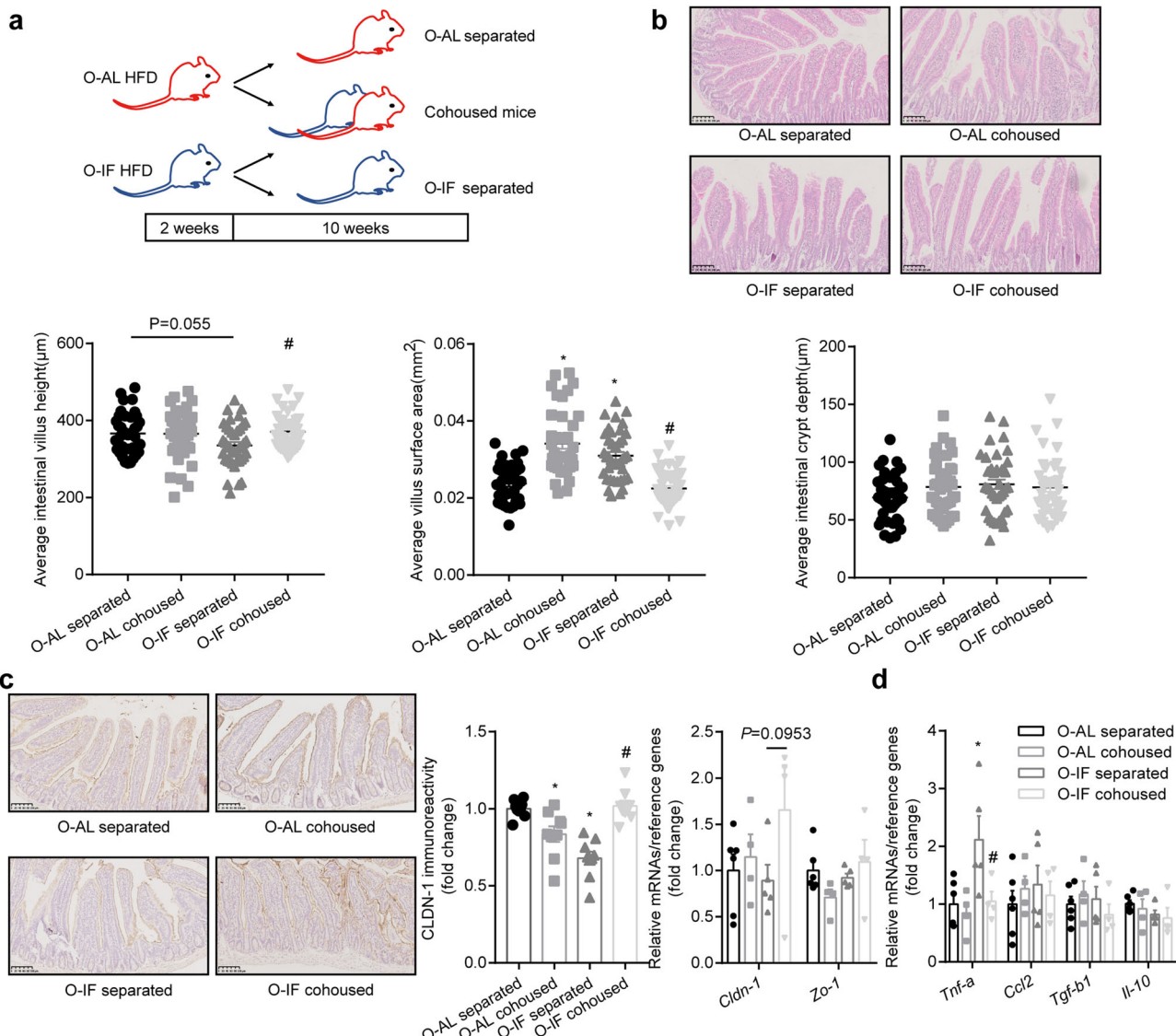

**Fig. 5 Offspring of maternal intermittent fasting mice co-housed with healthy mice restores intestinal barrier dysfunction. a** Experimental design. Six-week-old offspring of M-AL group and M-IF group were fed HFD for 2 weeks, then randomly selected and co-housed for 10-weeks. Results were expressed as mean ± SEM. *$P < 0.05$ vs. O-AL separated. #$P < 0.05$ vs. O-IF separated. $N = 6$ for O-AL separated, 4 for O-AL cohoused, 5 for O-IF separated, and 4 for O-IF cohoused. **b** Intestinal histomorphology. H&E staining of the small intestine and quantitative results of villus height, villus surface area, and crypt depth. **c** Immunoreactivity of CLDN-1 and mRNA levels of tight junction related genes (*Cldn-1* and *Zo-1*) determined by real-time quantitative PCR. **d** mRNA levels of inflammation related genes (*Tnf-a*, *Ccl2*, *Il-6* and *Tgf-b1*), which were determined by real-time quantitative PCR and normalized to the geometric mean value of reference genes (*Hprt*, *Rpl32* and *Tbp*).

the maternal IF offspring with healthy mice also rescued the impairment of intestinal barrier and disorder of glucose and lipid metabolism.

**Prolonged maternal intermittent fasting impairs intestinal barrier via suppression of beneficial microbiota in offspring.** Our present studies demonstrate that prolonged maternal intermittent fasting before pregnancy significantly reduces the diversity of intestinal microbiota in adult offspring. Substantial reduction in beneficial bacteria such as *Lactobacillus_intestinalis* leads to subsequent impairment in intestinal barrier characterized by increase in plasma LPS and inflammatory cytokine, and decrement in tight junction protein. Decrease in epithelial cell proliferation and concurrent increase in apoptosis also suggest an impairment in intestinal epithelial turnover. Our study thus extends the plasticity response of intestinal homeostasis to

maternal nutrition. However, it is worth of noting that we only measured a proportion of intestinal barrier functions and the interpretation of our findings should be cautioned before the complex trait of intestinal barrier function is extensively investigated. Previous studies have suggested that intestinal *Lactobacillu* (such as *L. reuteri*, *L. rhamnosus*, *Lactobacillus crispatus* and *L. plantarum*) is critical for the integrity of intestinal epithelial barrier by increasing cell-to-cell junctions and alleviating inflammation[26,27]. *Lactobacillus* (such as *Lactobacillus crispatus* and *Lactobacillus iners*) have been detected in the placental membranes and vagina in healthy, term pregnancies[28,29], suggesting that *Lactobacillus* may be one of the important links between dam and offspring. In line with these findings, our studies demonstrate that restoration of *Lactobacillus_intestinalis* or co-housing with healthy mice rescue the defect in intestinal barrier of maternal IF offspring. We thus propose that maternal

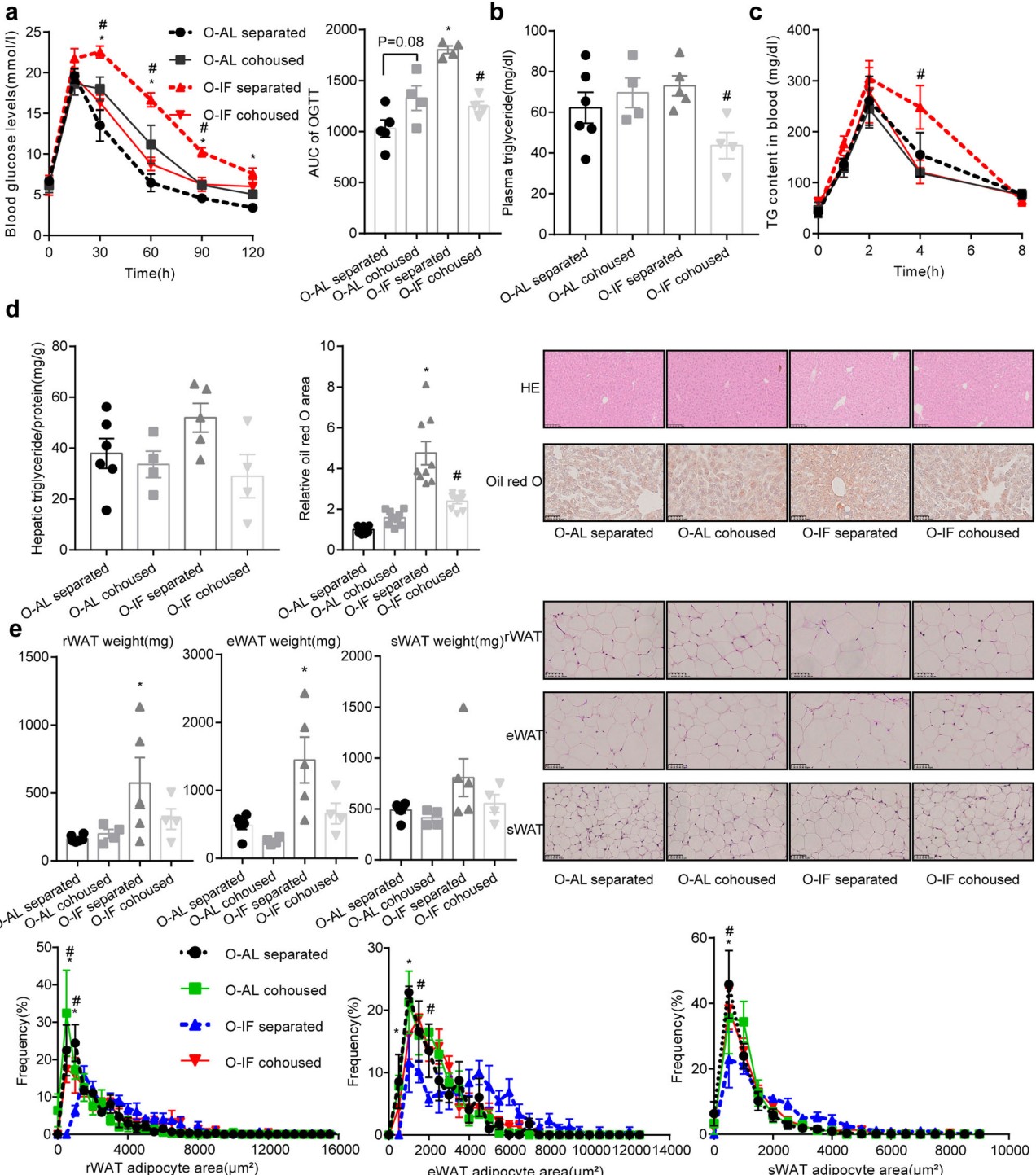

**Fig. 6 Maternal intermittent fasting offspring co-housed with healthy mice shows improvement in glucose and lipid metabolism.** O-AL and O-IF mice were co-housed for 10 weeks. Results were expressed as mean ± SEM. Two-way ANOVA was used for comparisons between multiple groups. *$P < 0.05$ vs. O-AL separated. #$P < 0.05$ vs. O-IF separated. **a** Glucose tolerance test and the area under curve. $N = 5$ for O-AL separated, 4 for O-AL cohoused, 4 for O-IF separated, and 4 for O-IF cohoused. **b** Plasma levels of triglyceride. $N = 6$ for O-AL separated, 4 for O-AL cohoused, 5 for O-IF separated, and 4 for O-IF cohoused. **c** Plasma levels of triglyceride after oral administration of olive oil. $N = 5$ for O-AL separated, 4 for O-AL cohoused, 4 for O-IF separated, and 4 for O-IF cohoused. **d** Triglyceride contents, oil red O and H&E staining of liver. $N = 6$ for O-AL separated, 4 for O-AL cohoused, 5 for O-IF separated, and 4 for O-IF cohoused. **e** Fat mass, H&E staining and adipocyte size of rWAT, eWAT, sWAT. $N = 6$ for O-AL separated, 4 for O-AL cohoused, 5 for O-IF separated, and 4 for O-IF cohoused.

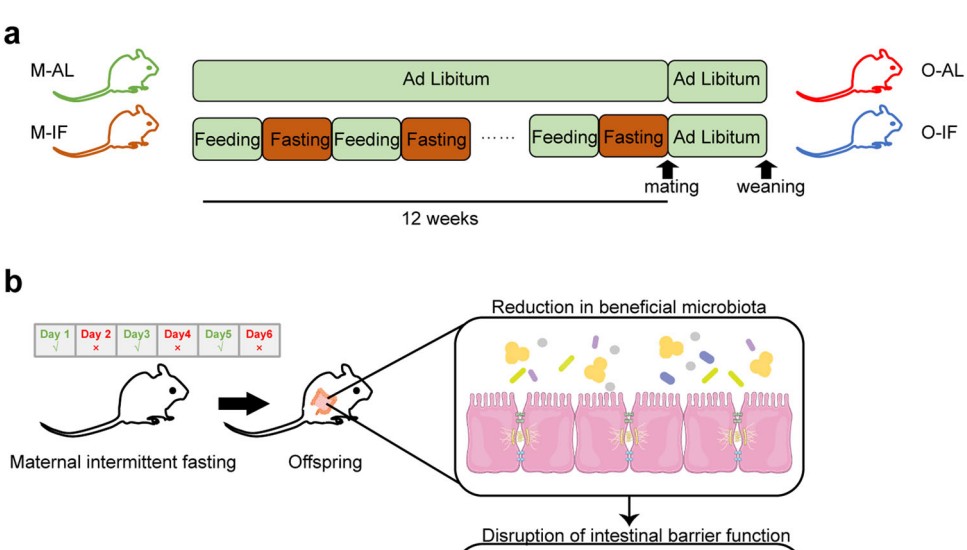

**Fig. 7 Graphic highlight of findings. a** An experiment scheme of maternal intermittent fasting. **b** Maternal intermittent fasting reduces beneficial microbiota in offspring intestine, leading to disruption of intestinal barrier function of offspring and subsequent disorders of glucose and lipid metabolism. This figure was partly generated using Servier Medical Art, provided by Servier, licensed under a Creative Commons Attribution 3.0 unported license.

intermittent fasting primes the offspring for the reduction in beneficial gut microbiota. Whether this is attributed to the decrement in the intergenerational transmission of *Lactobacillus* or adverse environment unfavorable for the growth of the beneficial bacteria remains unknown. Indeed, maternal nutrition or inflammation exposure during pregnancy have been reported to impact microbial transmission from mother to infant, leading to long-term consequences[30,31]. Further, previous studies have indicated that microbiota can inhabit the intrauterine environment. Bacteria has been detected in fetal membranes[32], umbilical cord blood[33], amniotic fluid[34] and placenta[35], suggesting that the establishment of the offspring microflora may initiate at the embryonic stage. In contrast to this concept, a series of studies has suggested that the utero is germ free[36,37]. In addition, previous studies in both mice and human have suggested that intermittent fasting have altered the mother's gut microbiota, leading to significant increase in *Lactobacillus*[38,39]. Our data reveals a significant reduction of *Lactobacillus* in offspring of IF dams. These observations do not support the concept that offspring inherits the gut microbiota colonies directly from the mother. Rather, maternal intermittent fasting may alter the offspring gut microbiota through a mechanism yet to be defined. Further well-designed animal and human studies are thus required to define the key maternal molecules signaling for reduction of beneficial intestinal microbiota in offspring.

**Metabolic consequence of intestinal microbiota dysbiosis induced by long-term maternal intermittent fasting in offspring.** The homeostasis of intestinal microbiota is crucial for the host health[40]. Our studies further support the concept that prolonged maternal intermittent fasting may induce intestinal microbiota dysbiosis in offspring with metabolic consequence. Unexpectedly, long-term maternal intermittent fasting reduces the beneficial intestinal bacteria with *Lactobacillus_intestinalis* as the dominant phylotypes. Restoration of *Lactobacillus_intestinalis* rescues the metabolic phenotypes in the offspring. Glucose

tolerance, adiposity and liver steatosis are all reduced. These observations suggest that reduction in beneficial bacteria such as *Lactobacillus_intestinalis* may contribute to the metabolic consequence of prolonged maternal intermittent fasting on the offspring. Consistently, the abundance of *L. intestinalis* has been reported to be negatively correlated with obesity and fat mass[41,42]. The beneficial metabolic effect of long-term treatment with *Lactobacillus* has been observed in both rats[43,44] and humans[45,46].

In summary, our study demonstrates that maternal intermittent fasting disrupts intestinal barrier leading to disorder of glucose and lipid metabolism. These effects occur through microbiota dysbiosis characterized by significant reduction in beneficial bacteria such as *Lactobacillus_intestinalis*. Our observations further support the concept that intestinal microbiota in offspring is vulnerable to maternal nutrition, and its homeostasis is critical for the integrity of intestinal barrier and metabolic homeostasis.

## Methods

**Animals and treatment**. All experiments were conducted in strict accordance with the Guide for the Care and Use of Laboratory Animals prepared by the National Academy of Sciences (NIH publication 86–23, revised 1985). Experimental protocols were approved by the Peking University Health Science Center. Four-week-old C57BL/6 J female mice ($n = 10$) and male mice (used for the mating, $n = 10$) were purchased from the Department of Laboratory Animal Science at Peking university Health Science Center. Animals were housed in the SPF-level environment with a standard environment ($22 \pm 2\,°C$, humidity at $50 \pm 15\%$) with 12 h light and 12 h dark cycle. Food and water were freely accessible except for the fasting experiment. At the end of the experiment, following tissue samples were harvested from offspring: plasma, intestine and its digesta, liver, retroperitoneal white adipose tissue (rWAT), epididymal white adipose tissue (eWAT) or parametrial white adipose tissue (pWAT), and subcutaneous white adipose tissue (sWAT).

Alternate-day maternal intermittent fasting protocol: 4-week-old female mice were subject to fasting every other day for 12 weeks, then these 16-week-old female mice were mated, and normal feeding resumed (M-IF group). Control female mice were fed *ad libitum* (M-AL group) (Fig. 7a).

Offspring feeding: 6-week-old male offspring of M-AL group (O-AL) and M-IF group (O-IF) were fed high fat diet (HFD, 60% fat, D12492; Research Diets) or normal chow diet (NCD, D12450H; Research Diets) for 12 weeks.

**Table 1 Primers for quantitative RT-PCR.**

| Genes | Upstream primer (5′-3′) | Downstream primer (5′-3′) |
|---|---|---|
| Cldn-1 | CTGGAAGATGATGAGGTGCAGAAG | CCACTAATGTCGCCAGACCTGAA |
| Ocln | TGAAAGTCCACCTCCTTACAGA | CCGGATAAAAAGAGTACGCTGG |
| Tjp | GCTTGCTGACCTACCCTGTG | CACTGCCAGACTGAGCTGAAT |
| Zo-1 | GAGGCTTCAGAACGAGGCTATT | CATGTCGGAGAGTAGAGGTTCGA |
| Tnf-a | CCAGACCCTCACACTCAGATC | CACTTGGTGGTTTGCTACGAC |
| Ccl2 | TAAAAACCTGGATCGGAACCAAA | GCATTAGCTTCAGATTTACGGGT |
| Il-6 | CTGCAAGAGACTTCCATCCAG | AGTGGTATAGACAGGTCTGTTGG |
| Il-17a | CCTCAGACTACCTCAACCG | CTCCCTCTTCAGGACCAG |
| Tgf-b1 | CTTCAATACGTCAGACATTCGGG | GTAACGCCAGGAATTGTTGCTA |
| Il-10 | CTTACTGACTGGCATGAGGATCA | GCAGCTCTAGGAGCATGTGG |
| Cd36 | AGATGACGTGGCAAAGAACAG | CCTTGGCTAGATAACGAACTCTG |
| Hprt | TCAGTCAACGGGGGACATAAA | GGGGCTGTACTGCTTAACCAG |
| Rpl32 | GAGCAACAAGAAAACCAAGCA | TGCACACAAGCCATCTACTCA |
| Tbp | ACCTTATGCTCAGGGCTTGG | GCCGTAAGGCATCATTGGAC |

*Lactobacillus_intestinalis* treatment: After fed HFD for 2 weeks, O-AL and O-IF male mice were daily gavaged with 200 μL of freshly prepared suspension of *Lactobacillus_intestinalis* at a dose of $2 \times 10^8$ CFU or PBS for 10 weeks (Fig. 3a).

Co-housing experiment: to allow for efficient transfer of microbes between O-AL and O-IF mice, O-Al and O-IF male mice were fed HFD for 2 weeks, then randomly co-housed for 10 weeks (Fig. 5a).

**Detection of LPS.** LPS in serum was measured using the Tachypleus Amebocyte Lysate (TAL) assay as previously described[47].

**AimPlex multiple immunoassays.** Intestinal cytokines were assayed in an AimPlex Platform employing mouse Th1/Th2/Th17 18-plex kit (C281118, Beijing QuantoBio Biotechnology Co., Ltd., Beijing, China) following the manufacturer's instructions. Results were normalized by total protein in the intestinal tissue extract.

**Culture of *Lactobacillus_intestinalis*.** *Lactobacillus_intestinalis* (ATCC49335) purchased from American Type Culture Collection (ATCC) were cultured in De Man, Rogosa and Sharpe (MRS) broth for 12 h at 37 °C in 5% $CO_2$, then centrifuged at $1000 \times g$ for 10 min at 20 °C, washed with PBS, and re-suspended in PBS at the concentration of $10^9$ CFU/ml.

**Gut microbiota analysis by 16 S rRNA gene sequencing.** Total genome DNA from samples harvested from ileum contents was extracted using CTAB/SDS method. Following 16 s rRNA V3-V4 region amplification, PCR products quantification and qualification, DNA library was constructed using the TruSeq® DNA PCR-Free Sample Preparation Kit. Sequencing libraries were generated using TruSeq® DNA PCR-Free Sample Preparation Kit (Illumina, USA) following manufacturer's recommendations and index codes were added. The library quality was assessed on the Qubit@ 2.0 Fluorometer (Thermo Scientific) and Agilent Bioanalyzer 2100 system. After the library qualified by Qubit and Q-PCR, NovaSeq6000 was used for sequencing and 250 bp paired-end reads were generated. Sequences with ≥97% similarity were assigned to the same OTUs. Shannon index was applied in analyzing complexity of species diversity, calculated with QIIME (Version 1.7.0) and displayed with R software (Version 2.15.3). LEfSe analysis LDA score >4 was shown.

**Oral glucose tolerance tests (OGTT) and insulin tolerance test (ITT)**
*OGTT.* Mice fasted for 16 h were orally gavaged with glucose at a dose of 3 g/kg body weight. Blood was collected from the incision at the tip of the tail at 0, 15, 30, 60, 90, and 120 min after glucose administration, and the glucose concentration was immediately measured.

*ITT.* Mice were fasted for 6 h before intraperitoneal insulin administration at a dose of 0.75U/kg body weight. Blood was collected from the incision at the tip of the tail at 0, 15, 30, 60, 90, and 120 min after glucose administration, and the glucose concentration was immediately measured.

**Oral lipid tolerance test (OLTT).** Mice fasted for 16 h were orally gavaged with olive oil (200 μL). Blood was collected from inner canthus at 0, 1, 2, 4, and 8 h after olive oil administration, and serum triglyceride were measured by colorimetry.

OGTT, ITT and OLTT tests were performed on the same mice. Ad libitum diets were resumed immediately after each test. Mice were allowed to recover for 5 days before next test.

**Tissue sample preparation, H&E, immunohistochemistry (IHC) and TUNEL staining.** The dissected tissues were fixed with 4% paraformaldehyde in PBS for 24 h at 4 °C and stored in 20% sucrose phosphate buffer. Samples were embedded in paraffin and sectioned at a 3 μm thickness. Intestine 5–10 cm distal to the pyloric sphincter was used for IHC and H&E staining. IHC and H&E were performed following general protocols. For IHC staining, each group contains three mice, three segments of intestine were selected and the immunoreactivity of CLDN-1 was quantified using Image J software. Anti-CLDN-1 was obtained from Abcam (Cambridge, MA). For H&E staining, each group contains 3-4 mice. Ten crypts or villus were randomly selected from each slide and analyzed using Image J software. TUNEL staining was performed according to the manufacturer's instructions (C1098, Beyotime, China).

**Oil red O staining and quantification.** Samples were embedded in OTC and frozen sections at 8 to10μm prepared. After gently washing with PBS, sections were fixed with 4% paraformaldehyde for 10 mins, then stained with freshly prepared Oil Red O working solution for 60 mins. After rinsing with distilled water, nuclei were counterstained with haematoxylin. Sections were rinsed with water and sealed with 90% glycerin. Quantitative analysis was performed by Image J software.

**Western blot and quantitative RT-PCR.** Intestinal protein was quantitated and loaded (25 μg protein/lane) onto an SDS-PAGE gel and transferred to a PVDF membrane. Membrane was blocked with 5% nonfat dry milk in TBST at room temperature for 1 h, then incubated with the primary antibody overnight at 4 °C. Anti-PCNA was obtained from ABclonal (Wuhan, China). Anti-β-actin was purchased from Cell Signaling Technology (Beverly, MA). IRDye-labeled secondary antibodies were used to detect specific reactions and visualized with the Odyssey infrared imaging system.

Intestinal RNA was isolated from tissues using Trizol, followed by reverse transcription. Quantitative real-time PCR was performed using SYBR green in Agilent AriaMx real-time PCR system. Table 1 shows the primer sequences involved in this study.

**Statistics and reproducibility.** Animal experiments were independently repeated at least two times with consistent results. Using Prism software for graphing and statistical analysis, the experimental results were shown as mean ± SEM. Statistical significance of differences between the groups was analyzed with a *t*-test or one/two-way ANOVA, followed by Tukey multiple post hoc analysis. $P < 0.05$ denotes statistical significance.

### Data availability

The source data behind the graphs in the paper can be found in Supplementary Data 1. Raw sequences have been deposited on NCBI public repository (Bioproject # PRJNA774493).

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

## Acknowledgements

This study was supported by National Natural Science Foundation of China (81730020, 81930015, 82070592 and 82270610).

## Author contributions

Y.L.: data acquisition, manuscript drafting; W.Y. and C.L.: animal breeding and data acquisition; Y.Y. and W.Z.: experiment design, manuscript revision, funding support; L.S., T.F., and Y.Z.: technical and reagents support. All other authors edited and approved the final manuscript.

## Competing interests

The authors declare no competing interests.
