## [Peer Review File · Communications Biology]

Reviewers' comments:

Reviewer #1 (Remarks to the Author):

The manuscript describes results from an experiment where intermittent fasting was compared to normal feeding of mice before breeding on offspring intestinal effects. The topic is of interest and the research question seems justified. My primary comments/suggestions are: 1. the authors claim they are measuring intestinal barrier function, yet intestinal barrier function is a complex trait and authors measured a small number that might be indicative of changes in barrier function. Therefore, the discussion needs to take this into account and more care is needed in the interpretation of the results. 2. It is difficult to determine how many mice were used in the experiments. This should be explicitly stated with information on body weight, etc. of the test animals. It seems that the number of observations is quite low. 3. The treatment design of the experiment is a 2*2 factorial and the data should be analyzed as such. The authors could examine if there is an interaction effect between maternal treatment and offspring diet. 4. The methods are largely lacking with very few details on how the experiments and analyses were conducted. 5. The treatment effects needs to be more clearly state (including p-values) in the results section. Also, statistical results are largely lacking from the figures. and 6. Sentence structure and grammar needs improvement.

Reviewer #2 (Remarks to the Author):

Methods section requires major revisions to include details of the experiments performed including IF protocol, detailed study designs for each experiment performed; protocols for OGGT, ITT, OLTT (how many animals per group, when was this performed, were the same mice used for all tests, multiple 16 hours fasting protocols; time to recover between tests, etc; which portion of the intestine was used for immunoassays: what time point was LPS analyzed (fasted/fed blood sample); methods for all histological assessments are missing and oil red O protocol/quantification not included; the lack of details prevents other researchers from reproducing this work. Statistical parameters used for microbiota analyses were not reported. Results section is difficult to follow when study designs or research questions are not clearly defined. Diet intake and growth rate of offspring should be reported. The impact of IF on maternal microbiota should have been reported or at least the rationale for why it was not included since it is a mechanism suggested in the introduction.

Reviewer 1:

The manuscript describes results from an experiment where intermittent fasting was compared to normal feeding of mice before breeding on offspring intestinal effects. The topic is of interest and the research question seems justified. My primary comments/suggestions are:

1. The authors claim they are measuring intestinal barrier function, yet intestinal barrier function is a complex trait and authors measured a small number that might be indicative of changes in barrier function. Therefore, the discussion needs to take this into account and more care is needed in the interpretation of the results.

Reply: Per the suggestion, we have addressed the limitation of our findings and revised the interpretation in lines 19-25, page 6 in the discussion section.

2. It is difficult to determine how many mice were used in the experiments. This should be explicitly stated with information on body weight, etc. of the test animals.

Reply: Information on the number of mice is now provided in figure legends. And body weight information is included in Fig S1a and Fig S2a.

3. The treatment design of the experiment is a 2*2 factorial and the data should be analyzed as such. The authors could examine if there is an interaction effect between maternal treatment and offspring diet.

Reply: Two-way ANOVA was used in our study. The interaction effect between maternal treatment and offspring diet is now included in lines 11-13, page 4.

4. The methods are largely lacking with very few details on how the experiments and analyses were conducted.

Reply: We have added the details on how the experiments and analysis were conducted.

5. The treatment effects needs to be more clearly state (including p-values) in the results section. Also, statistical results are largely lacking from the figures.

Reply: We have revised the results section accordingly.

6. Sentence structure and grammar needs improvement.

Reply: We have carefully revised the manuscript to improve sentence structure and grammar.

Reviewer 2:

1. Methods section requires major revisions to include details of the experiments performed including IF protocol, detailed study designs for each experiment

performed; protocols for OGTT, ITT, OLTT (how many animals per group, when was this performed, were the same mice used for all tests, multiple 16 hours fasting protocols; time to recover between tests, etc); which portion of the intestine was used for immunoassays: what time point was LPS analyzed (fasted/fed blood sample); methods for all histological assessments are missing and oil red O protocol/quantification not included, the lack of details prevents other researchers from reproducing this work.

Reply: The detail information is now provided.

IF protocol and study designs in lines 11-14, page 8: Alternate-day maternal intermittent fasting: four-week-old female mice were subject to fasting every other day for 12 weeks, then sixteen-week-old female mice returned to normal feeding at the time of mating (M-IF group). Control female mice were fed ad libitum (M-AL group) (Fig. 7a).

OGTT, ITT and OLTT in lines 27-29, page 9

OGTT, ITT and OLTT tests were performed in the same set of mice. ad libitum diets were resumed immediately after the experiment. Five days to recover between tests. The number of animals per group was included in figure legends.

Lines 31-34, page 9: 5-10 cm distal to the pyloric sphincter was used for intestine IHC and H&E staining.

Line 23, page 8: Serum LPS in fed mice was measured using the Tachypleus Amebocyte Lysate (TAL) assay.

Lines 34-38 page 9 and lines 1-7 page 10: Methods for all histological assessments and oil red O protocol/quantification were now included.

2. Statistical parameters used for microbiota analyses were not reported. Results section is difficult to follow when study designs or research questions are not clearly defined.

Reply: Per suggestion, this information is now included in line 11, page 15.

3. Diet intake and growth rate of offspring should be reported.

Reply: Food intake and growth rate (measured by body weight) of offspring are now included in Fig S2a.

4. The impact of IF on maternal microbiota should have been reported or at least the rationale for why it was not included since it is a mechanism suggested in the introduction.

Reply: We did not analyze the dam's gut microbiota because the impact of IF on gut microbiota has been extensively investigated. Previous studies in both mice and human have suggested that intermittent fasting alters the mother's gut microbiota, leading to significant increase in Lactobacillus^{1,2}. Our data reveals a significant reduction of Lactobacillus in offspring from IF dams. These observations do not support the concept that offspring inherits the gut microbiota colonies directly from the mother. Rather, maternal intermittent fasting may alter the offspring gut microbiota through a mechanism yet to be defined.

References:

- 1. Rinninella, E. et al. Gut Microbiota during Dietary Restrictions: New Insights in Non-Communicable Diseases. Microorganisms. 8, (2020).*
- 2. Cignarella, F. et al. Intermittent Fasting Confers Protection in CNS Autoimmunity by Altering the Gut Microbiota. Cell Metab. 27, 1222-1235 (2018).*